# A Robust Assay to Monitor Ataxin-3 Amyloid Fibril Assembly

**DOI:** 10.3390/cells11121969

**Published:** 2022-06-19

**Authors:** Francisco Figueiredo, Mónica Lopes-Marques, Bruno Almeida, Nena Matscheko, Pedro M. Martins, Alexandra Silva, Sandra Macedo-Ribeiro

**Affiliations:** 1Instituto de Investigação e Inovação em Saúde (i3S), Universidade do Porto, 4200-135 Porto, Portugal; francisco.figueiredo@ibmc.up.pt (F.F.); mmarques@i3s.up.pt (M.L.-M.); pmartins@ibmc.up.pt (P.M.M.); 2Instituto de Biologia Molecular e Celular (IBMC), Universidade do Porto, 4200-135 Porto, Portugal; 3International Iberian Nanotechnology Laboratory (INL), 4715-330 Braga, Portugal; 4Instituto de Ciências Biomédicas Abel Salazar (ICBAS), Universidade do Porto, 4050-313 Porto, Portugal; 5Department of Biology, Faculty of Sciences, University of Porto, 4169-007 Porto, Portugal; 6Life and Health Sciences Research Institute (ICVS), School of Medicine, University of Minho, 4710-057 Braga, Portugal; brunoalmeida@med.uminho.pt; 7ICVS/3B’s—PT Government Associate Laboratory, 4710-057 Braga, Portugal; 8Dynamic Biosensors GmbH, 82152 Martinsried, Germany; matscheko@dynamic-biosensors.com

**Keywords:** thioflavin-T, polyglutamine expansion, reproducibility, ubiquitin, self-association rates, equilibrium dissociation constant, switchSENSE

## Abstract

Spinocerebellar ataxia type 3 (SCA3) is caused by the expansion of a glutamine repeat in the protein ataxin-3, which is deposited as intracellular aggregates in affected brain regions. Despite the controversial role of ataxin-3 amyloid structures in SCA3 pathology, the identification of molecules with the capacity to prevent aberrant self-assembly and stabilize functional conformation(s) of ataxin-3 is a key to the development of therapeutic solutions. Amyloid-specific kinetic assays are routinely used to measure rates of protein self-assembly in vitro and are employed during screening for fibrillation inhibitors. The high tendency of ataxin-3 to assemble into oligomeric structures implies that minor changes in experimental conditions can modify ataxin-3 amyloid assembly kinetics. Here, we determine the self-association rates of ataxin-3 and present a detailed study of the aggregation of normal and pathogenic ataxin-3, highlighting the experimental conditions that should be considered when implementing and validating ataxin-3 amyloid progress curves in different settings and in the presence of ataxin-3 interactors. This assay provides a unique and robust platform to screen for modulators of the first steps of ataxin-3 aggregation—a starting point for further studies with cell and animal models of SCA3.

## 1. Introduction

Ataxin-3 is a modular protein that contains a globular Josephin domain and a flexible C-terminal tail, which includes two or three ubiquitin interaction motifs (UIMs) and a polyglutamine (polyQ) stretch. Expansion of the polyQ tract above a threshold of 45–55 residues leads to ataxin-3 self-assembly and triggers spinocerebellar ataxia type 3 (SCA3), also known as Machado–Joseph disease, a highly incapacitating autosomal-dominant neurodegenerative disorder. Deposition of amyloid-like fibrillar aggregates containing the mutant polyQ-expanded protein in neurons represents a characteristic hallmark of SCA3 and of other polyQ expansion diseases [1,2]. Although the precise neurotoxic culprit is widely debated, protein aggregation is a major contributing factor to the underlying neurodegeneration, as pathogenic ataxin-3 has been shown to accumulate in intracellular deposits at the affected brain regions, such as the cerebellum, brain stem, and spinal cord [3,4]. This perception is strengthened by the observation that increasing the length of the polyQ tract enhances protein aggregation propensity in vitro, in correlation with an upsurge in disease severity [5,6,7]. Despite nearly 30 years of intensive research, which started with the identification of the causative gene and continued with the discovery of ataxin-3 function(s) and its aggregation mechanisms, SCA3 remains an incurable disease justifying the urgent need for disease-modifying therapies [8]. Several strategies have been employed to therapeutically target polyglutamine disorders based on the prevention of the misfolding and aggregation of proteins containing expanded polyglutamine tracts (reviewed by Minakawa and Nagai [9]). Given the morphological heterogeneity of the aggregates and the transient nature of the aggregation intermediates, identifying protein aggregation inhibitors with neuroprotective properties is a challenge and requires a thorough knowledge of the self-assembly pathways of the target proteins.

Previous studies have shown that ataxin-3 assembly into amyloid fibrils follows a multi-step pathway involving multiple domains and at least two well-defined aggregation steps. The first step is mediated by an aggregation-prone region located in the Josephin domain [10,11,12,13]. This aggregation-prone region (^73^GFFSIQVISNALKVWGLELILFNS^96^) leads to the formation of sodium dodecyl sulfate (SDS)-soluble protofibrils both in non-expanded and expanded ataxin-3. The second step, leading to the formation of large, stable, and SDS-resistant amyloid-like fibrils, is polyQ-dependent and exclusive to expanded ataxin-3 [11,14,15]. The complexity of the ataxin-3 aggregation pathway renders the characterization of the self-assembly mechanisms and its modulation by point mutations, interacting partners, or aggregation inhibitors particularly challenging. We have noticed that small changes in buffer conditions significantly modify ataxin-3 aggregation kinetics in vitro, altering both the oligomerization processes and the balance between ordered fibrillar structures and amorphous aggregation. In this context, studying ataxin-3 aggregation mechanisms in vitro under controlled experimental conditions can provide instrumental information to evaluate how different factors and molecules modulate its assembly into ordered fibrillar structures—a starting point for further studies with cell and animal models of SCA3.

Robust and reproducible aggregation assays are necessary to explore ataxin-3 self-assembly and screen for aggregation inhibitors that could lead to the development of disease-modifying therapies for SCA3 [16]. Here, we present detailed assays developed to monitor the aggregation of wild-type (Atx3 13Q) and polyQ-expanded ataxin-3 (Atx3 77Q) using a miniaturized thioflavin-T (ThT) assay in combination with transmission electron microscopy (TEM) visualization to characterize aggregate morphology [17]. Using this optimized assay, we evaluated the impact of a broad range of conditions on ataxin-3 aggregation in vitro, including ionic strength, pH, the presence of detergents, and molecular crowders. Furthermore, we show how ataxin-3-interacting peptides and proteins, such as polyQ binding peptide 1 (QBP1) and linear ubiquitin chains (previously reported to modulate ataxin-3 aggregation [18,19,20,21,22,23]), modulate ataxin-3 amyloid assembly kinetics and fibril morphology.

## 2. Materials and Methods

### 2.1. Expression Plasmids

The plasmid constructs used in this work are listed in Table 1. Sequence details are provided in the Appendix A. The cDNAs coding for human normal (13Q) and polyQ-expanded (77Q) ataxin-3 (Atx3) (isoform 2, UniProt accession: P54252-2) with three ubiquitin interaction motifs (UIMs) were cloned into the pDEST17 Gateway vector (Life Technologies, Carlsbad, CA, USA) as previously described [10]. The plasmids were further modified to introduce a tobacco etch virus (TEV) protease cleavage site downstream of the hexahistidine tag [11]. To prepare the constructs with codons optimized for improved expression of repeated polyQ sequences in E. coli, two synthetic constructs (Appendix A) were designed and purchased from GenScript (Piscataway, NJ, USA) and inserted between the restriction sites Ppu10I and BsrGI of the original *ATXN3* gene sequence cloned in pDEST17. Site-directed mutagenesis (Appendix A) performed with the QuikChange II Site-Directed Mutagenesis Kit (Agilent Technologies, Santa Clara, CA, USA) was used to produce Atx3 13Q I77K Q78K W87K and the Atx3 77Q R388G variant (Single Nucleotide Polymorphism Database locus accession code: rs12895357, UniProt VAR_013689). The plasmid for the expression of untagged *Homo sapiens* polyubiquitin with 3 ubiquitin repeats in tandem (pET23a-HsUBB) was a gift from Prof. Jorge Azevedo [23] (Addgene plasmid #69562).

### 2.2. Ataxin-3 Expression and Purification

Ataxin-3 variants and truncated constructs were expressed and purified as previously described [10] with minor modifications. Briefly, *E. coli* BL21(DE3)-SI cells (Life Technologies) were transformed with pDEST17 plasmids carrying the various ataxin-3 variants and truncated constructs and plated on LB Agar without NaCl (LBON) plates supplemented with 100 µg mL^−1^ ampicillin. A pre-inoculum was prepared with 4-5 isolated colonies in 150 mL LBON medium supplemented with 100 µg mL^−1^ ampicillin, and cells were grown overnight at 37 °C, 180 rpm. The following day, 10 mL of pre-inoculum was added to 500 mL of LBON medium supplemented with 100 µg mL^−1^ ampicillin and 0.4% (*w/v*) glucose and grown at 37 °C, 180 rpm, until the OD_600nm_ reached 0.8. The culture was then cooled to 30 °C and protein expression was induced with 300 mM NaCl. After 3 h of expression, the cells were harvested by centrifugation and resuspended in buffer A (20 mM sodium phosphate pH 7.5, 500 mM NaCl, 2.5% (*v/v*) glycerol, 20 mM imidazole) containing 100 µg L^−1^ lysozyme.

For ataxin-3 purification, the cells were disrupted by gentle stirring for 1 h on ice in the presence of 0.02 mg mL^−1^ DNase, 0.02 mg mL^−1^ RNase, 1 mM MgCl_2_, and 1 mM phenylmethylsulfonyl fluoride (PMSF—from a 1 M stock solution in ethanol). The supernatant obtained after centrifugation was loaded onto a Ni^2+^-charged IMAC HiTrap column (GE Healthcare Life Sciences, Piscataway, NJ, USA) pre-equilibrated in buffer A and eluted with a series of imidazole steps (50 mM, 250 mM, and 500 mM). EDTA was added to a final concentration of 1 mM to eluted fractions containing ataxin-3 to reduce proteolytic cleavage. The 250 mM imidazole fraction was applied to a HiPrep 26/60 Sephacryl S-300 HR column (GE Healthcare Life Sciences) pre-equilibrated in protein storage buffer (20 mM sodium phosphate pH 7.5, 150 mM NaCl, 5% (*v/v*) glycerol, 2 mM EDTA, 1 mM DTT). After SDS–PAGE analysis, fractions from this column corresponding to pure ataxin-3 isoforms were pooled and concentrated in an Amicon Ultra-15 centrifugal filter unit (10 kDa, Millipore, Burlington, MA, USA) to 10–20 mg mL^−1^, frozen in liquid nitrogen, and stored at −80 °C. Protein concentration was determined by measuring the absorbance at 280 nm using the protein’s extinction coefficients, as detailed in Appendix A.

Immediately before each aggregation assay, purified ataxin-3 aliquots were applied to a Superose 12 ^®^ 10/300 GL column (GE Healthcare Life Sciences) pre-equilibrated in aggregation buffer (20 mM sodium phosphate pH 7.5, 150 mM NaCl, 1 mM DTT or 20 mM HEPES pH 7.5, 150 mM NaCl, 1 mM DTT) and 0.4 mL fractions were collected. Protein elution was monitored at 280 nm, and the protein concentration of the fraction corresponding to monomeric protein was determined from the absorbance at 280 nm (Appendix A).

### 2.3. HsUBB Expression and Purification

Homo sapiens polyubiquitin containing 3 ubiquitin units (HsUBB) was expressed and purified as previously described [23] with minor modifications. Briefly, *E. coli* BL21 (DE3) cells (Life Technologies) were transformed with pET23a-HsUBB and plated on LB Agar medium supplemented with 100 µg mL^−1^ ampicillin. A pre-inoculum was prepared from 4–5 isolated colonies in LB medium supplemented with 100 µg mL^−1^ ampicillin and grown overnight at 37 °C, 180 rpm. The pre-inoculum was added to LB medium supplemented with 100 µg mL^−1^ ampicillin and grown at 37 °C, 180 rpm, until the OD_600nm_ reached 0.8. HsUBB expression was induced with 0.5 mM IPTG for 3 h at 37 °C. The cells were harvested by centrifugation and resuspended in 50 mM Tris–HCl pH 8.5, 1 mM EDTA, 0.5 mM DTT, 0.25 mg mL^−1^ PMSF. Cells were lysed by sonication and the lysate was clarified by centrifugation (34,957 g, 45 min, 4 °C—Sorvall ST 40R—Rotor JA25.50). The cell lysate was applied to a 6 mL Resource Q ion-exchange column (GE Healthcare Life Sciences) and the flow-through was concentrated and injected onto a Superdex 75 (GE Healthcare Life Sciences) column equilibrated in 50 mM Tris–HCl pH 8.5, 150 mM NaCl, 10% (*v/v*) glycerol, 1 mM DTT. The peak corresponding to pure HsUBB was concentrated in an Amicon Ultra-15 centrifugal filter unit (10 kDa, Millipore) and stored at −80 °C. The protein concentration was determined by measuring the absorbance at 280 nm using the extinction coefficient of 5,960 M^−1^cm^−1^. Before the aggregation assay with ataxin-3, HsUBB was re-purified as described in Section 2.2 using a Superose 12 ^®^ 10/300 GL column (GE Healthcare Life Sciences) pre-equilibrated with the ataxin-3 aggregation buffer.

### 2.4. Thioflavin-T Aggregation Assay

Ataxin-3 amyloid formation was monitored by following the increase in ThT fluorescence at 480 nm (440 nm excitation) on a fluorimeter. For this study, two different fluorimeters were used: the FluoDia T70 microplate fluorimeter (Photon Technology International, Edison NJ, USA) using Thermowell 96-Well Polycarbonate PCR Microplates (Costar, Washington, DC, USA) and the CHAMELEON V (HIDEX) plate reader using 384-well microplates (low flange, black, flat bottom, polystyrene; Corning, Corning, NY, USA). Samples (50 μL) of 5 µM ataxin-3 in aggregation buffer containing 30 μM ThT and different concentrations of the tested compounds (as detailed below), were incubated at 37 °C, and ThT fluorescence was measured every 30 min for 60 h. To prevent evaporation, each well was covered with 20 μL paraffin oil.

### 2.5. Transmission Electron Microscopy

For visualization of protein fibrils/aggregates by TEM, endpoint protein samples used in the ThT assay were diluted five-fold in water and adsorbed onto glow-discharged, carbon-coated films supported on 300-mesh nickel grids and negatively stained with 1% (*w/v*) uranyl acetate using a protocol adapted from Rames and collaborators [24]. Grids were visualized using a JEM-1400 (JEOL) TEM at an accelerating voltage of 80 kV.

### 2.6. Filter Retardation Assay

To detect Atx3 77Q mature SDS-resistant fibrils, samples (5 µL) collected from each well at the endpoint of the ThT aggregation assay were diluted in 200 µL of TBS (50 mM Tris–HCl pH 7.5, 150 mM NaCl) supplemented with 2% (*w/v*) SDS and boiled for 5 min. Using a Bio-Dot^®^ SF microfiltration apparatus (Bio-Rad, Hercules, CA, USA), the samples were filtered through a cellulose acetate membrane (0.2 µm, Whatman, Buckinghamshire, UK) pre-equilibrated in TBS, and the membrane was washed twice with TBS supplemented with 0.1% (*w/v*) SDS. Next, the membrane was removed from the apparatus and blocked with TBS supplemented with 5% (*w/v*) non-fat dry milk for 1 h at room temperature (RT). The Atx3 77Q mature SDS-resistant fibrils retained in the membrane were probed with monoclonal mouse anti-Atx3 clone 1H9 (Millipore) antibody 1:10,000 overnight at 4 °C, incubated with anti-mouse antibody (Sigma, Burlington, MA, USA) 1:10,000 for 1 h at RT, and detected using Amersham ECL Prime Western blotting chemiluminescent detection reagent (Cityva, Marlborough, MA, USA).

### 2.7. Native Gel Assay

Samples of 40 µM Atx3 13Q or Atx3 77Q were incubated with HsUBB at different molar ratios (1:0.5; 1:1; 1:2 and 1:3) in 20 mM HEPES pH 7.5, 150 mM NaCl, 5% (*v/v*) glycerol, 2 mM EDTA, 1 mM DTT, for 1 h on ice. Proteins were loaded and separated in an 8% native polyacrylamide gel electrophoresis (PAGE) gel (240 mM Tris–HCl pH 9.5, 8% (*v/v*) acrylamide, 0.1% (*w/v*) APS, 0.1% (*v/v*) TEMED), in a Mini-PROTEAN Tetra cell (Bio-Rad) at 100 V, and stained by incubation with BlueSafe (NZYTech).

### 2.8. Ataxin-3 13Q and Ataxin-3 JD Oligomerization Kinetics Measured via SwitchSENSE

Measurements were set up in the switchBUILD software with “His-tag capture” as the immobilization method and performed in static measurement mode on a DRX device on MPC-48-2-R1 biochips (both Dynamic Biosensors GmbH (DBS), Munich, Germany). The system was primed with running buffer (phosphate-buffered saline (PBS) pH 7.5, 1 mM TCEP) and the biochip was functionalized with 200 nM NTA3-functionalized cNL-B48 DNA (His-tag capture kit, order number CK-TN-1-B48, DBS) in hybridization buffer (10 mM sodium phosphate pH 7.4, 40 mM NaCl, 0.05% (*v/v*) Tween 20, 50 µM EDTA, 50 µM EGTA) for 10 min. The NTA3 group was stripped and activated by injection of EDTA and loading solution. Capture of 500 nM His-tagged Atx3 13Q or the Atx3 JD domain, diluted in running buffer and stored at 10 °C in the autosampler until automatic pickup, was performed for 2 min at a flow rate of 20 µL min^−1^ at 37 °C. Association of Atx3 13Q or Atx3 JD domain diluted in running buffer to indicated concentrations, as well as subsequent dissociation in running buffer, was performed at 37 °C, 100 and 50 µL min^−1^, respectively. Samples were measured in consecutive rounds with increasing analyte concentrations and NTA regeneration in between. Data were analysed with the switchANALYSIS software kinetics tool. Data were normalized to the baseline of the association, Atx3 13Q data were referenced with a 0 M run, and global mono-exponential fits were calculated by least-squares residual determination.

## 3. Results

### 3.1. Development of a Miniaturized Ataxin-3 Aggregation Assay

Ataxin-3 self-assembly has a complex mechanism, with oligomeric and amyloid pathways taking place simultaneously [17,25]. The high tendency of ataxin-3 to assemble into high-molecular weight oligomeric structures [10], concomitantly with the occurrence of two parallel ataxin-3 oligomerisation pathways on- and off-route to the formation of amyloid fibrils, implies that a tight control of the multiple variables and aggregation buffer components is critical to the development of reproducible assays to monitor protein aggregation [16]. We have previously established a miniaturized assay to follow the amyloid assembly of Atx3 13Q [16,17]. In that assay, freshly re-purified Atx3 13Q (5 µM) was incubated at 37 °C in phosphate buffer and amyloid formation was monitored by ThT binding in a final volume of 50 µL. Under those previously established conditions, the lag phase of aggregation was longer than 20 h and often varied between 30 and 40 h [17]. The long lag phase of ataxin-3 amyloid assembly in phosphate buffer, combined with the molecular stochasticity of the primary nucleation, accounts for the large variations in the lag phase duration that we observed during the aggregation assays under these conditions. In contrast, the lag phase for Atx3 13Q aggregation was decreased by ~20 h when sodium phosphate (as in [16,17]) was replaced with HEPES in the aggregation solution (Figure 1A). The aggregation assay using HEPES buffer was very robust, as shown by the high reproducibility observed in 192 replicates setup in a 386-well plate using 3 µM Atx3 13Q (Appendix A), suggesting that this assay can be used for medium- and high-throughput screening for ataxin-3 aggregation inhibitors.

Next, we evaluated the behaviour of ataxin-3 variants and truncated constructs (Figure 1B) in aggregation solutions buffered with HEPES. The purified proteins (Appendix A), stored at −80 °C, were thawed and re-purified by size exclusion chromatography in the selected buffer before the aggregation assay (Figure 1D). This step is critical to remove the glycerol used in the ataxin-3 storage buffer, which has a strong effect on the aggregation kinetics (see below), and to eliminate putative aggregates formed during freezing/thawing, which could modify the nucleation process and compromise assay reproducibility. A complete protocol including all the steps for ataxin-3 production and aggregation analysis is presented in the Appendix A. One relevant factor to ensure high-quality protein is the utilization of codon-optimized nucleotide sequences to express ataxin-3 with various polyQ tract sizes, particularly important for producing high-purity Atx3 77Q. Our results showed that ataxin-3 self-assembles into amyloid-like structures with fibrillar morphology independently of the polyQ tract (Figure 1E,H), similar to what had been previously observed in phosphate-buffered aggregation solutions [17]. In addition, SDS–PAGE analysis of ataxin-3 at the endpoint of aggregation (~60 h, 37 °C) showed that no relevant protein degradation occurred under the established assay conditions (Figure 1F).

Previous data showed that both Atx3 77Q and Atx3 13Q aggregate, with the globular JD playing a central role in the first steps of ataxin-3 self-assembly [10,13,26,27]. To quantify Atx3 JD and Atx3 13Q self-association kinetics, we used electrically switchable nanolever (switchSENSE) technology [28,29]. SwitchSENSE is a biophysical characterization method ideally suited to kinetic binding analysis of challenging molecules, such as aggregation-prone proteins. The low surface density of the immobilized protein of interest allows a controlled induction of the oligomerization processes upon analyte injection in the liquid phase. The nanolevers were functionalized with 500 nM Atx3 13Q or JD and binding of unlabeled JD or Atx3 13Q to the surface-tethered proteins allowed the measurement of JD and Atx3 13Q self-association kinetics. The association and dissociation curves were measured for analyte concentrations between 3.75 and 15 µM. The equilibrium dissociation constants (K_D_) for Atx3 13Q and JD self-association (Figure 1G) were in the µM range, in good agreement with the K_D_ of 4.6 µM estimated from our biophysical model for Atx3 13Q aggregation [17]. Although the association rate constants (k_on_) were similar for both constructs, the isolated JD dissociated faster, showing that the C-terminal tail of ataxin-3 stabilizes its self-association, in good agreement with the differences observed in the ThT aggregation kinetics (Figure 1E).

Interestingly, the truncated variant lacking the polyQ segment and the C-terminal UIM3 (Atx3 D1) had the same amyloid assembly kinetics as the full-length ataxin-3, highlighting the relevance of the region bridging the JD and the polyQ for self-assembly into amyloid-like fibrils [18,30]. In contrast to previously published data [15,31], the aggregation kinetics for the non-expanded Atx3 13Q and disease-related Atx3 77Q fully overlapped. This result was obtained consistently when the proteins were freshly re-purified, immediately before the aggregation assays. This step removes minute amounts of aggregated species, most frequently contaminating freshly thawed Atx3 77Q monomeric fractions (Appendix A), which might act as nucleation seeds and accelerate aggregation of pathogenic ataxin-3. Analysis of the fibril morphologies by negative staining transmission electron microscopy (TEM; Figure 1H) showed that only Atx3 77Q assembled into mature fibrillar agglomerates [27], a process that apparently does not interfere with the maximum ThT signal (Figure 1E). This indicates that monitoring ataxin-3 aggregation by ThT binding kinetics only reflects the first aggregation step that is reliant on JD self-assembly. This assay may be used to evaluate the effects of mutations or compare the modulatory roles of various buffer components and ataxin-3-interacting molecules on ataxin-3 early aggregation processes, as shown below.

### 3.2. Analysis of the Aggregation Kinetics of Natural Ataxin-3 Variants

An intragenic polymorphism in the *ATXN3* gene (rs12895357, UniProt variant VAR_013689) was observed in the population, with one variant containing a GGG codon, translated to glycine downstream of the CAG repeats, and another variant containing a CGG codon, translated to arginine. This polymorphic position was used to study SCA3 haplotype and origins [32,33,34]. Taking into consideration the data available in the 1000 Genomes Project, Phase 3 [35], which gathered genomic information from 4973 healthy individuals, the variant containing the glycine residue after the polyQ repeat was prevalent in all populations (Appendix A). The variant containing arginine at the equivalent position was also found in all populations, being more prevalent in the East Asian population and was originally linked with repeat expansion in Japanese SCA3 patients [36].

Our Atx3 13Q construct contains a glycine residue after the polyQ sequence, while the Atx3 77Q construct has an arginine in the equivalent position (Appendix A). We compared the aggregation kinetics of Atx3 77Q and Atx3 77Q R388G (Figure 2) to evaluate the potential impact of the nature of the residue at this position upon expanded ataxin-3 aggregation. The results showed that the presence of a glycine or an arginine immediately after the polyQ tract did not alter the ThT fluorescence curves (Figure 2A), as they monitored the first step of ataxin-3 aggregation. Additionally, no differences were observed in the Atx3 77Q or Atx3 77Q R388G fibril morphologies (Figure 2B), both of which matured into SDS-resistant fibrillar clusters (Figure 2C). Therefore, we decided to use our original Atx3 77Q construct to explore the influence of various buffer components on ataxin-3 aggregation kinetics.

### 3.3. The Influence of the Buffer Components on Ataxin-3 Aggregation

Several studies have shown the impact of ionic strength on the aggregation propensity of various amyloid-forming proteins [37,38,39,40,41,42,43]. To evaluate the influence of ionic strength on ataxin-3 self-assembly, we monitored the effect of varying the NaCl concentration between 50 and 300 mM on the aggregation kinetics. In the selected buffer conditions, varying the NaCl concentration had no major effect on the formation of ThT-positive amyloid-like species for both expanded and non-expanded ataxin-3 (Figure 3A,B). However, large variability in the maximum ThT fluorescence in the different replicates was noted for the non-expanded Atx3 13Q in the presence of 300 mM NaCl (Figure 3A). This variability possibly resulted from the presence of air bubbles in this particular experiment, as replicates of this assay did not show a reduction of ThT signal (Appendix A). No relevant differences were found in the morphology of the fibrils at the endpoint of the ThT aggregation assay (~60 h) as observed by TEM analysis (Figure 3C). Similarly, no differences in ataxin-3 aggregation kinetics were observed when the pH of the HEPES buffer was varied between 7.0 and 8.5 (Appendix A).

We next tested the effect of glycerol on ataxin-3 aggregation. Glycerol is frequently used to prevent protein aggregation [44,45] and was added to the ataxin-3 purification and storage buffers [10,16,17]. We evaluated the effect of varying the glycerol concentration on ataxin-3 aggregation (Figure 4). In the presence of 0.5% or 1% (*v/v*) glycerol, the aggregation lag phase was elongated, both for polyQ-expanded and non-expanded ataxin-3 (Figure 4A,B). The presence of 5% (*v/v*) glycerol increased the aggregation lag phase and diminished the maximum ThT fluorescence for both ataxin-3 variants (Figure 4A,B). Accordingly, the morphological analysis of the endpoint samples by TEM showed a clear decrease in the size and number of Atx3 13Q protofibrils and Atx3 77Q mature fibrils in the presence of 5% (*v/v*) glycerol (Figure 4C).

The effects of other widely used buffer additives, such as (i) the chelating agent ethylenediamine tetraacetic acid (EDTA), (ii) the protease inhibitor phenylmethylsulfonyl fluoride (PMSF), (iii) the preservative sodium azide, and (iv) the solvents ethanol and dimethyl sulfoxide (DMSO), were also tested. All the conditions tested had no significant effect on the aggregation kinetics of both Atx3 13Q (Appendix A) or Atx3 77Q (Appendix A).

### 3.4. The Influence of Detergents on Ataxin-3 Aggregation

Detergents can be used to perturb hydrophobic protein–protein interactions and prevent unwanted protein aggregation. To evaluate the effect of detergents on ataxin-3 aggregation, we tested SDS and the milder biological detergents Triton X-100 and Tween 20.

SDS had an immediate effect on ThT fluorescence, suggesting that in the assay conditions this surfactant completely abrogated ataxin-3 aggregation (Figure 5A,D). A possible fluorescence quenching effect was observed in the presence of 1 mM or 5 mM SDS, where the ThT signal decreased to values below the initial zero. This effect was not as evident when SDS was added to ThT alone. TEM analysis detected no protofibrils or mature fibrils at the endpoint of Atx3 13Q or Atx3 77Q aggregation in the presence of SDS (Figure 5G). Previous studies had shown that 5 mM SDS enhanced the α-helical content of the JD, stabilizing ataxin-3 and interfering with both JD-mediated and polyQ expansion-triggered aggregation steps [46]. In contrast to our results, these previous studies found that 1 mM SDS increased ataxin-3 aggregation, using an assay with 6-fold higher ataxin-3 concentrations. We hypothesize that the higher SDS:ataxin-3 ratio in our assay might explain the divergent results observed with 1 mM SDS and that the anti-amyloid assembly effects of SDS might have been caused by specific interactions with ataxin-3 that prevented early fibril assembly events.

The effect of Triton X-100 and Tween 20 on ataxin-3 aggregation was evaluated using five different concentrations (0.15, 0.3, 0.8, 1, and 5 mM). Both detergents led to a sharp increase in the ThT signal at the onset of ataxin-3 aggregation (Figure 5B,C,E,F). As increasing concentrations of Triton X-100 (Figure 5B,E) or Tween 20 (Figure 5C,F) were used, the maximum value of ThT fluorescence decreased. Ataxin-3 aggregation curves in the presence of 5 mM Triton X-100 or Tween 20 showed lower ThT fluorescence values, which may suggest a strong anti-aggregation effect. The critical micellar concentrations (CMCs) of Triton X-100 and Tween 20 in water are ~0.2 mM and ~0.06 mM, respectively. Although these values may vary with the specific buffer conditions and temperature, the decrease in ataxin-3 aggregation with the increasing concentration of these non-ionic detergents could be correlated with ataxin-3 stabilization by interaction with the detergent micelles. However, a fluorescence quenching effect was observed when 5 mM Tween 20 or 5 mM Triton X-100 were added to ThT alone (Figure 5B,C,E,F), suggesting that the aggregation inhibition effects are likely artefactual [47]. TEM analysis at the endpoint of aggregation also revealed that the number and size of Atx3 13Q protofibrils were similar to those in the control condition (Figure 5G). Atx3 77Q was also able to form mature fibrils in the presence of these detergents (Figure 5G).

### 3.5. The Influence of Molecular Crowders on Ataxin-3 Aggregation

Molecular crowders are widely used in aggregation assays to mimic the high macromolecular concentrations within biological cells and predict the effects on protein aggregation [48]. For this reason, we studied the effect of well-known molecular crowders, such as dextran and polyethylene glycol (PEG), on the aggregation profiles of ataxin-3 (Figure 6).

We tested the effect of 10 and 20% (*w/v*) dextran, a commonly used crowding agent [48], on ataxin-3 aggregation. Dextran shortened the duration of the lag phase and decreased the maximum ThT fluorescence values of both ataxin-3 variants (Figure 6A,D). The effect of 20% (*w/v*) dextran was more pronounced on Atx3 77Q (Figure 6D), resulting in an inhibition of ataxin-3 amyloid-like aggregation, which was consistent with the absence of long mature fibrils (Figure 6G). Nevertheless, minor amounts of short protofibrils (Atx3 13Q) and amorphous aggregates (Atx3 77Q) were still observed (Figure 6G).

The addition of PEG 5000 or PEG 10,000 to ataxin-3 aggregation assays resulted in a concentration-dependent reduction of the maximum ThT fluorescence (Figure 6B,C,E,F), suggesting that PEG was able to interfere with the formation of ataxin-3 amyloid-like structures. However, the addition of 20% (*w/v*) PEG 5000 or PEG 10,000 inhibited the ataxin-3 aggregation process of both non-expanded and polyQ-expanded ataxin-3, as suggested by the absence of ThT fluorescence (Figure 6B,C,E,F). This effect was confirmed by the absence of fibrillar amyloid-like structures as observed by TEM (Figure 6G).

These results showed that dextran, PEG 5000, and PEG 10,000 were strong inhibitors of ataxin-3 amyloid assembly acting on the first stage of the aggregation pathway and consequently inhibiting the formation of Atx3 13Q protofibrils and mature Atx3 77Q fibrils.

### 3.6. The Influence of Ataxin-3 Interactors on Ataxin-3 Aggregation

Several studies have reported the interaction of ataxin-3 with ubiquitin and polyubiquitin chains [49,50,51,52,53]. Two ubiquitin binding sites have been identified in the JD, one of them overlapping with the aggregation-prone sequence, and the addition of ubiquitin has also been reported to decrease JD aggregation [54]. In agreement with this, mutation of the second ubiquitin binding site of ataxin-3, coincident with the aggregation-prone segment in the JD, affected the first JD-mediated aggregation step (Appendix A). We evaluated the interaction of Atx3 13Q and Atx3 77Q with a linear tri-ubiquitin chain (HsUBB) by native gel electrophoresis and the results suggested that both variants interact weakly with HsUBB (Figure 7A). A ThT aggregation assay of Atx3 13Q and Atx3 77Q in the presence of different molar ratios of HsUBB (1:2 and 1:5) showed that HsUBB did not change the aggregation kinetics of non-expanded ataxin-3 but had a slight impact on the maximum ThT fluorescence of Atx3 77Q (Figure 7B,C). At the highest concentration used, HsUBB reduced the number of Atx3 13Q protofibrils and the overall size of Atx3 77Q mature fibrils, as evaluated by TEM (Figure 7D). Analysis of the final products of Atx3 77Q aggregation by filter retardation assay revealed a slight decrease in the amount of Atx3 77Q SDS-resistant mature fibrils at the molar ratio of 1:5 (Figure 7E), suggesting a role for this linear ubiquitin chain in the modulation of pathogenic ataxin-3 aggregation.

The polyglutamine-binding peptide 1 (QBP1) is an 11-residue sequence that has been shown to inhibit the formation of mature SDS-resistant fibrils of polyQ-expanded ataxin-3 [15,18]. QBP1 binds to monomeric ataxin-3, between the JD domain and the UIM1 (residues 182–221), inhibiting the polyQ-dependent aggregation stage [18]. As expected, in our experiments, QBP1 did not prevent the formation of ataxin-3 protofibrils (Figure 8). For all QBP1:ataxin-3 molar ratios tested, we obtained coincident ThT aggregation curves for both ataxin-3 variants (Figure 8A,B). Samples of Atx3 13Q and Atx3 77Q in the presence of QBP1 at a molar ratio of 1:10 (ataxin-3:QBP1) were collected and analysed by TEM (Figure 8C). The results confirmed that the presence of QBP1 did not prevent the formation of Atx3 13Q protofibrils but that it did abolish the maturation of the Atx3 77Q fibrils (Figure 8C), as previously described [18].

## 4. Discussion

Several studies have shown that both normal and polyQ-expanded ataxin-3 are able to self-assemble into amyloid-like fibrils in vitro, opening new routes to study how various molecules modulate self-association and to discover anti-aggregation agents. Here, we present an optimized and robust assay to study the effect of distinct factors on the first step of ataxin-3 aggregation, following the guidelines that ensure assay reproducibility [16,55]. This assay requires low concentrations of protein (5 µM) and can be used to study both normal and polyQ-expanded ataxin-3. To ensure high reproducibility, ataxin-3 and/or its interacting partners are always re-purified by size-exclusion chromatography in the selected aggregation buffer immediately before setting up the aggregation assay. A detailed protocol for ataxin-3 production and aggregation studies is also included as ancillary information, along with the relevant data concerning the sequences of all the ataxin-3 constructs used in this work. Using this approach, we investigated how several external factors can affect protein aggregation in vitro and assessed the aggregation behaviour of ataxin-3 variants commonly found in the human population.

### 4.1. The Relevance of the Buffer System for Ataxin-3 Aggregation

Phosphate is considered a biologically relevant buffer, frequently used for studying amyloid formation in vitro. Our results show that the buffer system used in the self-assembly assays has an important impact on ThT-monitored ataxin-3 amyloid formation, as previously seen for Aβ(1–40) [56]. In particular, replacing sodium phosphate with HEPES induced an 80% reduction in the duration of the lag phase of the ataxin-3 amyloid progress curves and increased assay reproducibility, without affecting the morphology of the endpoint fibrils visualized by TEM. It is unclear why the two buffer ions significantly alter the lag phase, but this could be the result of weak interactions with unique ataxin-3 sequence motifs triggering or stabilizing the rate-limiting assembly of the first aggregation nuclei. This is an interesting question worth exploring in future research.

### 4.2. The Role of Additives on Ataxin-3 Aggregation

In our assay conditions, varying the NaCl concentration between 50 and 200 mM, or the pH between 7.0 and 8.5, did not strongly affect ataxin-3 amyloid assembly nor the morphology of the endpoint fibrils. In contrast, glycerol, an additive regularly used in ataxin-3 purification, had a major role in decreasing ataxin-3 fibril assembly, even at concentrations as low as 0.5% (*v/v*). Therefore, the presence of glycerol is an important parameter to define when designing the aggregation protocol. In contrast, EDTA (2-5 mM), PMSF (1 mM), sodium azide (0.05% (*w/v*)), ethanol (1% (*v/v*), and DMSO (0.05% (*v/v*)) did not affect the ataxin-3 amyloid progress curves.

### 4.3. The Effect of Detergents on Ataxin-3 Aggregation Monitored by ThT

The behaviour of the ThT curves for ataxin-3 amyloid assembly in the presence of anionic and non-ionic detergents is worth discussing in detail. Although ThT is an amyloid-specific dye commonly used to monitor amyloid formation kinetics in real time, particular care should be taken in the interpretation of the experimental results [16,57]. When SDS, Triton X-100, and Tween 20 were included in the aggregation assay, a decrease in the fluorescence signal of ThT was observed with increasing concentrations of the detergents. However, a sharp decrease in ThT fluorescence, observed in the control experiments without ataxin-3, suggested that these results should be interpreted with caution and confirmed by complementary techniques. In fact, TEM analysis showed that the anionic detergent SDS was able to abolish ataxin-3 assembly into amyloid fibrils, but the non-ionic detergents did not affect ataxin-3 amyloid formation.

### 4.4. Molecular Crowders and Ataxin-3 Aggregation

The intracellular environment is very complex. To mimic its complexity and study how crowding could impact ataxin-3 aggregation, we used different crowding agents. It is known that the effect of crowders on amyloid proteins varies with the concentration and type of crowder used [48,58,59,60]. Recently, it has been shown that huntingtin aggregation was affected by the presence of dextran, Ficoll, and PEG 20,000 increasing the heterogeneity of the huntingtin non-fibrillar aggregate species formed [58]. All the crowding agents tested (dextran, PEG 5000, and PEG 10,000) decreased the ThT signal in a concentration-dependent manner and abolished ataxin-3 fibril formation, as confirmed by TEM analysis. Macromolecular crowders are generally associated with a solubility-decreasing effect caused by the exclusion of protein molecules from the physical volume occupied by the crowding agents [61]. Nevertheless, when macromolecular crowders establish attractive interactions with the protein, solubility-enhancing effects may prevail over volume exclusion [62]. Since the studied crowding agents thermodynamically and kinetically inhibited ataxin-3 aggregation, we conclude that the enthalpic effects prevailed over the entropic ones.

### 4.5. The Effect of Ataxin-3-Interacting Proteins in Amyloid Assembly

Ataxin-3 is a deubiquitinase that preferentially cleaves K63-linked chains of four or more ubiquitin moieties [53,63]. Previous studies have shown that the surface areas involved in ubiquitin binding include the aggregation-prone regions in JD and that incubation with monomeric ubiquitin delayed its aggregation [49,54]. Since it had been suggested that native protein interactions may deter the self-assembly of aggregation-prone proteins [64], we tested the effect of HsUBB, a linear tri-ubiquitin chain whose structure is a close mimic of the extended conformation adopted by K63-linked ubiquitin chains [65], on ataxin-3 aggregation. Our results showed that in the tested experimental conditions, HsUBB does not interfere with ataxin-3 amyloid fibril assembly. However, for Atx3 77Q, a decrease in the maximum ThT fluorescence was observed at the highest molar ratio (1:5), suggesting that this may be an avenue of research worth pursuing.

QBP1 is an 11-residue peptide that can inhibit polyQ aggregation both in vitro and in vivo [18,19,20,21]. QBP1 is very efficient at inhibiting the formation of mature SDS-resistant fibrils without interfering with the first step of ataxin-3 aggregation. Our studies corroborated this and allowed us to verify that ThT binding curves are a good tool for evaluating modulators of the JD-mediated step of ataxin-3 aggregation, although TEM and filter retardation analysis should be complementarily used to monitor the expanded polyQ-dependent aggregation step.

## 5. Conclusions

In conclusion, we have developed a standard optimized assay to study the first step of ataxin-3 aggregation. We showed that small amounts of additives, in particular glycerol, can affect the aggregation process and that different control experiments are required to identify what type of effect is revealed by ThT fluorescence analysis. Further, the relevance of complementary studies to monitor fibril morphology by negative staining electron microscopy to obtain an integrated view of the effect of different molecules on amyloid fibril assembly and maturation has been highlighted. Several independent studies have shown that individuals with the same number of glutamine repeats have different ages of disease onset and varying symptoms [66,67,68,69,70,71]. These studies emphasize that polyQ expansion is not the only factor underlying the development of SCA3 and that other physiological and external factors play a role in the manifestation of the disease. This optimized aggregation assay can be used to study the effect of various modulators of disease progression on ataxin-3 self-assembly and to discover new ataxin-3 aggregation inhibitors for future therapy development.

## Figures and Tables

**Figure 1 cells-11-01969-f001:**
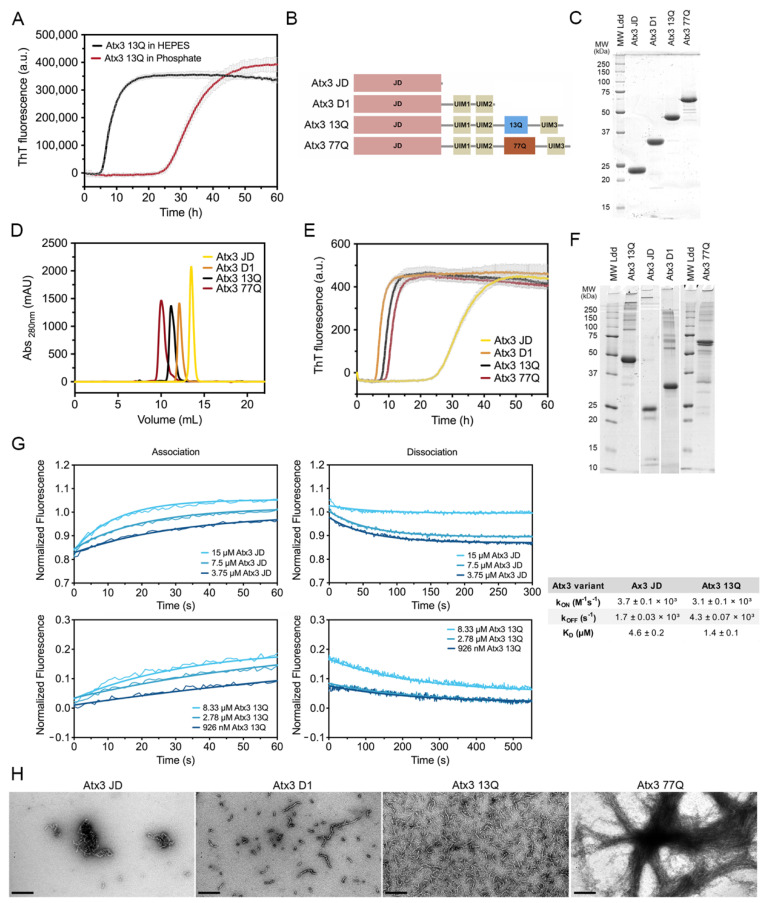
Ataxin-3 amyloid assembly assay. (**A**) ThT assay performed to monitor the formation of Atx3 13Q amyloid species in two distinct buffers, HEPES vs. phosphate buffer. Curves represent the means and standard deviations of five replicates for each condition (FluoDia T70 plate reader). (**B**) Schematic representation of the ataxin-3 constructs used in this work. (**C**) SDS–PAGE of freshly thawed ataxin-3 constructs prior to re-purification; MW Ldd– molecular weight markers. (**D**) Size exclusion chromatography of re-purified ataxin-3 isoforms, where no aggregates are detectable. (**E**) ThT assay performed to measure the formation of amyloid species for different ataxin-3 constructs in HEPES buffer. Curves represent the means and standard deviations of five replicates for each condition (CHAMELEON V plate reader). (**F**) SDS–PAGE of ataxin-3 constructs at the end of the ThT aggregation assay. (**G**) Self-assembly kinetics of Atx3 JD and Atx3 13Q determined using switchSENSE technology and derived rate and equilibrium constants. (**H**) TEM images after negative staining of ThT assay endpoint samples (60 h, 37 °C) of all ataxin-3 constructs. Scale bars correspond to 200 nm.

**Figure 2 cells-11-01969-f002:**
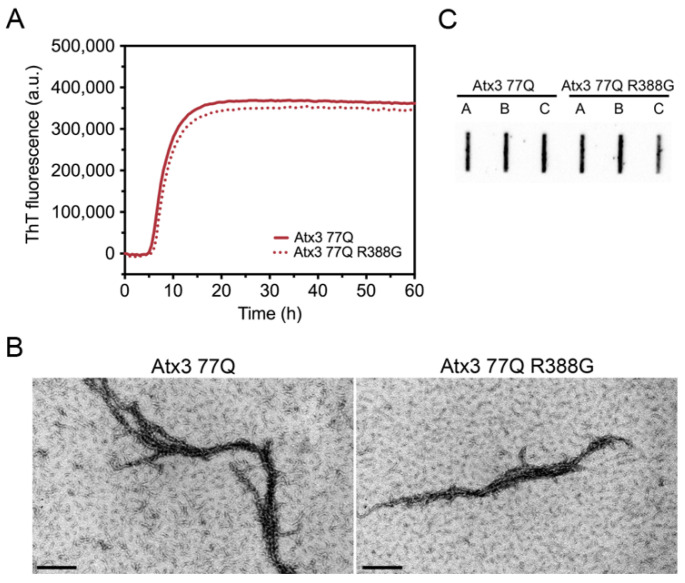
Effect of ataxin-3 variant rs12895357 on aggregation kinetics. (**A**) ThT assay performed to measure the formation of amyloid-like species in Atx3 77Q and Atx3 77Q R388G. Curves represent the means and standard deviations of five replicates for each condition (FluoDia T70 plate reader). (**B**) TEM images after negative staining of ThT assay endpoint samples (60 h, 37 °C) of both ataxin-3 variants. Scale bars correspond to 200 nm. (**C**) Filter retardation assay monitored by immunostaining with mouse anti-ataxin-3 1H9 of aggregation endpoints (60 h, 37 °C) from Atx3 77Q and Atx3 77Q R388G; the labels A, B, and C are replicates.

**Figure 3 cells-11-01969-f003:**
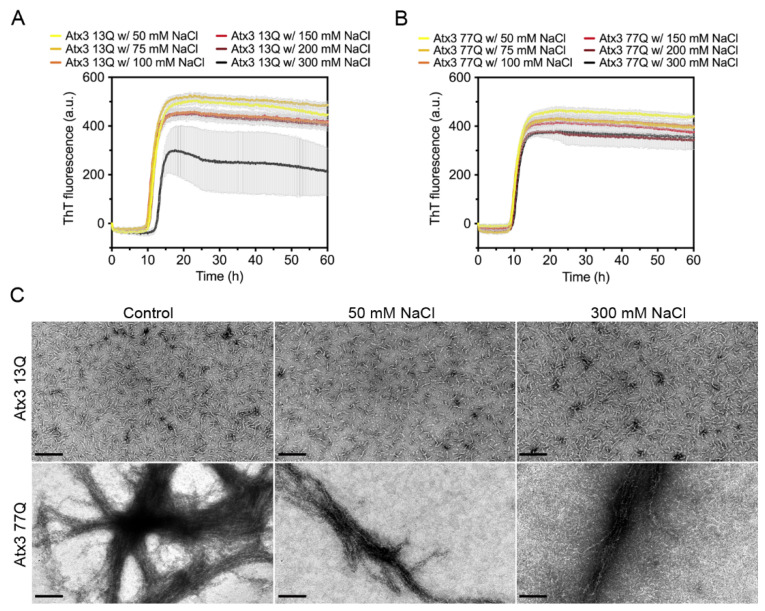
Effect of NaCl on ataxin-3 aggregation kinetics. ThT assay performed to measure the formation of amyloid-like species in (**A**) Atx3 13Q and (**B**) Atx3 77Q in the presence of 50, 75, 100, 150 (control), 200, and 300 mM NaCl. Curves represent the means and standard deviations of five replicates for each condition (CHAMELEON V plate reader). (**C**) TEM images after negative staining of ThT assay endpoint samples (60 h, 37 °C) of Atx3 13Q and Atx3 77Q in the presence of 150 (control), 50, and 300 mM NaCl. Scale bars correspond to 200 nm.

**Figure 4 cells-11-01969-f004:**
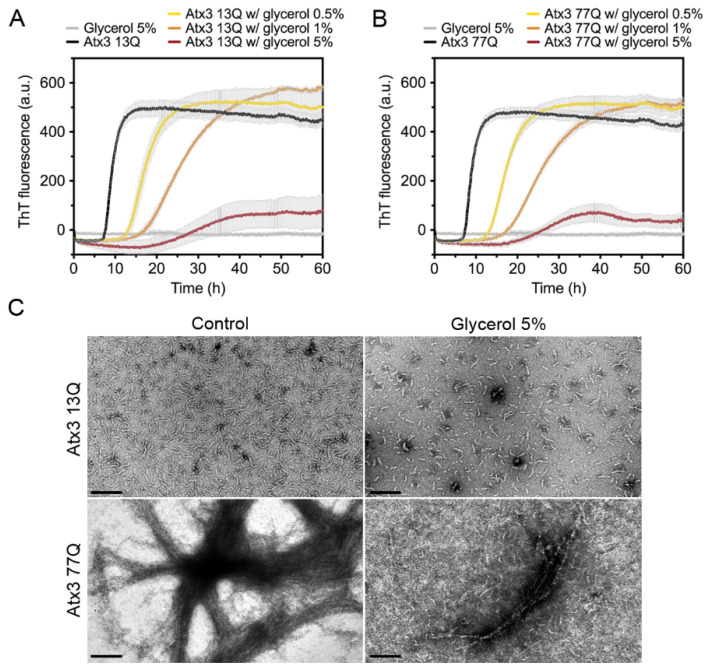
Effect of glycerol on ataxin-3 aggregation kinetics. ThT assay performed to measure the formation of amyloid-like species in (**A**) Atx3 13Q and (**B**) Atx3 77Q in the presence of 0.5, 1, and 5% (*v/v*) glycerol. Curves represent the means and standard deviations of five replicates for each condition (CHAMELEON V plate reader). (**C**) TEM images after negative staining of ThT assay endpoint samples (60 h, 37 °C) of Atx3 13Q and Atx3 77Q in the absence (control) and in the presence of 5% (*v/v*) glycerol. Scale bars correspond to 200 nm.

**Figure 5 cells-11-01969-f005:**
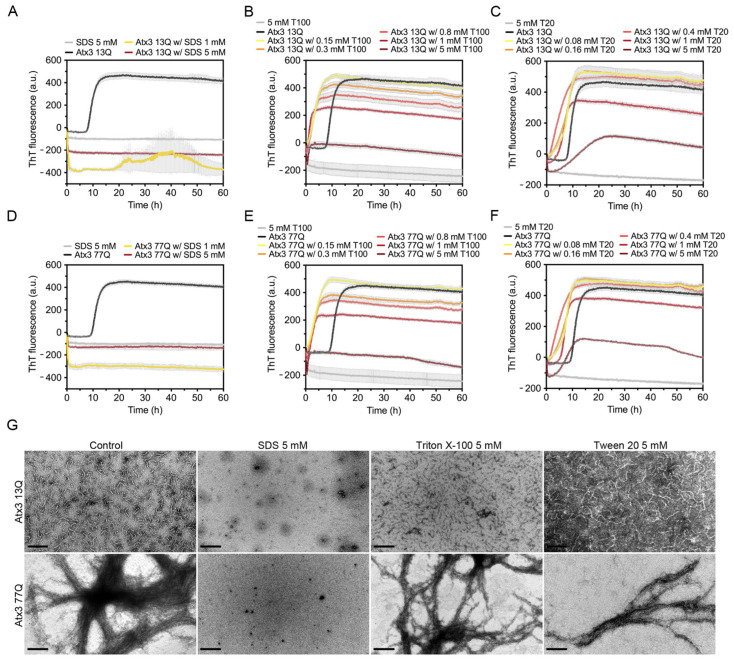
Effect of SDS, Triton X-100, and Tween 20 on ataxin-3 aggregation kinetics. ThT assay to measure the formation of amyloid-like species in (**A**–**C**) Atx3 13Q and (**D**–**F**) Atx3 77Q in the presence of different concentrations of SDS, Triton X-100 (T100), and Tween 20 (T20). Curves represent the means and standard deviations of five replicates for each condition (CHAMELEON V plate reader). (**G**) TEM images after negative staining of ThT assay endpoint samples (60 h, 37 °C) of Atx3 13Q and Atx3 77Q in the presence of 5 mM SDS, Triton X-100, or Tween 20, in comparison with the control condition in the absence of these additives. Scale bars correspond to 200 nm.

**Figure 6 cells-11-01969-f006:**
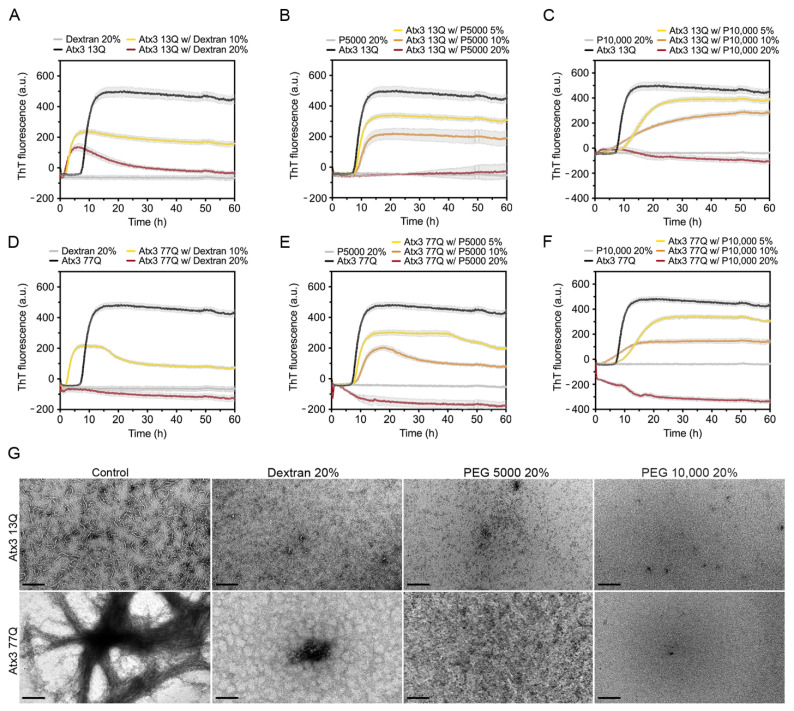
Effects of dextran, PEG 5000, and PEG 10,000 on ataxin-3 aggregation kinetics. ThT assay performed to measure the formation of amyloid-like species in (**A**–**C**) Atx3 13Q and (**D**–**F**) Atx3 77Q in the presence of different concentrations of dextran, PEG 5000, and PEG 10,000. Curves represent the means and standard deviations of five replicates for each condition (CHAMELEON V plate reader). (**G**) TEM images after negative staining of ThT assay endpoint samples (60 h, 37 °C) of Atx3 13Q and Atx3 77Q in the presence of Dextran, PEG 5000 (P5000), and PEG 10,000 (P10,000), compared to the control condition in the absence of additives. Scale bars correspond to 200 nm.

**Figure 7 cells-11-01969-f007:**
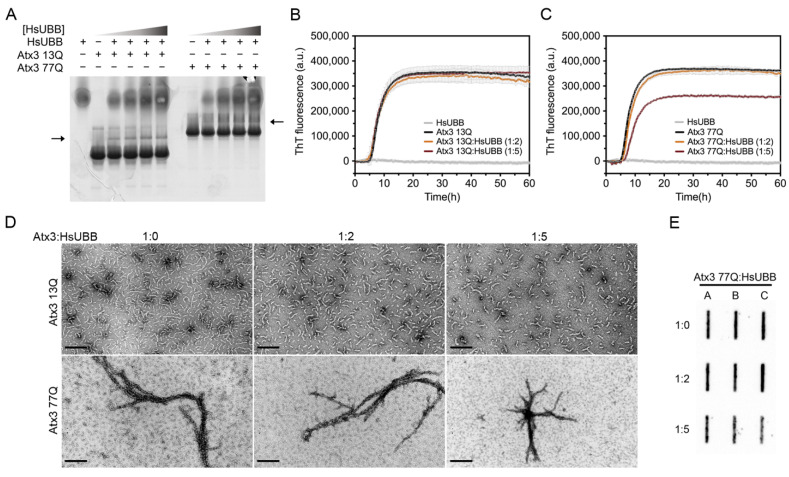
Effect of HsUBB on ataxin-3 aggregation kinetics. (**A**) Native gel electrophoresis of Atx3 13Q and Atx3 77Q with different ratios of ataxin-3:HsUBB, suggesting a weak interaction between ataxin-3 and tri-ubiquitin. ThT assay performed to measure the formation of amyloid-like species of (**B**) Atx3 13Q and (**C**) Atx3 77Q in the presence of 10 and 25 μM HsUBB. Curves represent the means and standard deviations of five replicates for each condition (FluoDia T70 plate reader). HsUBB was re-purified by size exclusion chromatography in ataxin-3 aggregation buffer prior to the aggregation assay. (**D**) TEM images after negative staining of ThT assay endpoint samples (60 h, 37 °C) of Atx3 13Q and Atx3 77Q in the presence of different concentrations of HsUBB. Scale bars correspond to 200 nm. (**E**) Filter retardation assay of aggregation endpoint samples (60 h, 37 °C) from the Atx3 77Q aggregation assay in the presence of HsUBB; the labels A, B, and C are replicates.

**Figure 8 cells-11-01969-f008:**
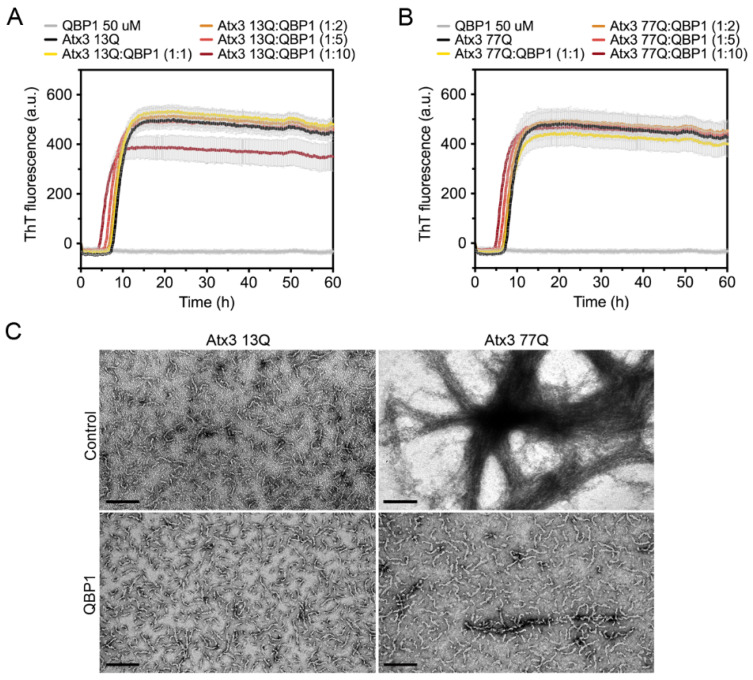
Effect of QBP1 on ataxin-3 aggregation kinetics and fibril maturation. ThT assay performed to measure the formation of amyloid-like species of (**A**) Atx3 13Q and (**B**) Atx3 77Q in the presence of 5, 10, 25, and 50 µM QBP1 in ataxin-3 aggregation buffer. Curves represent the means and standard deviations of five replicates for each condition (CHAMELEON V plate reader). (**C**) TEM images after negative staining of ThT assay endpoint samples (60 h, 37 °C) of Atx3 13Q and Atx3 77Q in the presence of QBP1 at a molar ratio of 1:10 (ataxin-3:QBP1). Scale bars correspond to 200 nm.

**Table 1 cells-11-01969-t001:** Summary of the constructs used in the present work and adopted nomenclature. For sequence details, see Appendix A.

Construct Name	Protein Expressed	Original Reference	Addgene ID
Atx3 JD	Ataxin-3 Josephin domain	[10,11]	184247
Atx3 D1	Ataxin-3 JD + UIM1-2	[22]	184246
Atx3 13Q	Ataxin-3 (13Q) with a glycine residue after the polyQ tract	This work	184248
Atx3 13Q I77K Q78Q W87K	Ataxin-3 (13Q) with a triple mutation on ubiquitin-binding site 2	This work	185908
Atx3 77Q	PolyQ-expanded ataxin-3 (77Q) with an arginine residue after the polyQ tract (UniProt natural variant VAR_013689)	[15]	184249
Atx3 77Q R388G	PolyQ-expanded ataxin-3 (77Q) with a glycine residue after the polyQ tract	This work	184251
HsUBB	*Homo sapiens* linear tri-ubiquitin chain	[23]	69562

## Data Availability

The datasets generated and/or analyzed during the current study are available from the corresponding authors upon request. All plasmids used in this work are deposited at Addgene.

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
