# Peer review of "A Robust Assay to Monitor Ataxin-3 Amyloid Fibril Assembly"

_cells, 2022, doi:10.3390/cells11121969_

Round 1

Reviewer 1 Report

The manuscript entitled “Development of a robust assay to monitor ataxin-3 amyloid fibril assembly” examines the aggregation of ataxin-3 (variants and truncated forms) under numerous experimental conditions in vitro.  

Specifically, transmission electron microscopy and thioflavin-T kinetic assays were used to study the impact of NaCl, glycerol, buffer additives, detergents, molecular crowders, and interactors on the aggregation kinetics of ataxin-3. 

Overall, this work highlights the factors that should be considered when studying the aggregation process of ataxin-3, as well as the need for more than one method to monitor the aggregation kinetics of a protein.

Although the manuscript is well written and of interest to anyone who wants to study ataxin-3, its title is quite misleading.  The authors claim that they developed an optimized protocol for medium- and high-throughput screening for modulators of the first steps of ataxin-3 aggregation, but they do not provide a comparison between the conventional and their optimized protocols, by showing which steps of the conventional protocols have been optimized, and by explaining why the protocol is better than the conventional ones.  Additionally, they should explain how this protocol ensures reproducibility. 

These results are undoubtfully useful for researchers who want to further study the aggregation process of ataxin-3, but I do not think that they can be considered a protocol, at least the way they are presented.

In general, I strongly advise the authors to change the title of the manuscript or to focus more on the description of the “protocol” in the main text.

Minor comments:

1.      The authors should re-read their supplementary material as there are some grammatical errors (e.g., was be made).  Additionally, some figures are missing (Figure S1 and Figure S5).

2.     The scale bars of the TEM figures are barely visible.  The authors should change either the color of the bars or the weight of the stroke.  Also, they do not need to write the value of the bar on the figure, since it can be barely seen.

3.     The authors should check the numbering of the heading.  The heading 3.6 should be followed by the heading 3.6.1 and not 3.1.1 (line 475, page 15).

4.     The authors should explain why someone should use the switchSENSE method and why did they use it only once in their experiments.

Author Response

Major comments

Point 1

Although the manuscript is well written and of interest to anyone who wants to study ataxin-3, its title is quite misleading.  The authors claim that they developed an optimized protocol for medium- and high-throughput screening for modulators of the first steps of ataxin-3 aggregation, but they do not provide a comparison between the conventional and their optimized protocols, by showing which steps of the conventional protocols have been optimized, and by explaining why the protocol is better than the conventional ones.  Additionally, they should explain how this protocol ensures reproducibility. 

These results are undoubtfully useful for researchers who want to further study the aggregation process of ataxin-3, but I do not think that they can be considered a protocol, at least the way they are presented.

In general, I strongly advise the authors to change the title of the manuscript or to focus more on the description of the “protocol” in the main text.”

Response 1: We thank the reviewer for the time dedicated to read our manuscript and for the insightful comments and suggestions.

In line with the reviewer’s comment we have removed all references to the “development of optimized protocol” in the abstract and in the main text of the manuscript. Additionally, the title was altered to “A robust assay to monitor ataxin-3 amyloid fibril assembly” and we detailed the relevant steps that we consider critical to ensure highly reproducible ataxin-3 aggregation assays (please see lines 228-242 and 519-524 of the revised manuscript). In particular, we included additional Atx3 13Q aggregation data with 192 replicates, showing that the assay described in this work is robust and suitable for medium- and high-throughput screening of ataxin-3 aggregation inhibitors (please see Supplementary Figure S1 and lines 239-242 in the revised manuscript). The different results presented in sections 3.2 to 3.6 of the manuscript further vouch for the high quality and reproducibility of the data: the lag phases were similar in the various ataxin-3 amyloid progress curves obtained, even when using different batches of purified proteins and at least two different plate readers.

Comparison with previously published studies reporting amyloid assembly progress curves of ataxin-3 is challenging, because important experimental information is often lacking and the exact sequences of the constructs used are not mentioned in some of the publications. Therefore, we chose to provide a brief comparison with our previously published assays and explain how the assay was modified to enhance reproducibility (lines 228-232, page 6; lines 519-524, page 17). We are confident that the protocols for ataxin-3 production and aggregation assays included in the Supplementary information of the current manuscript, along with the disclosure of the amino acid sequences of the proteins used in this work, will contribute towards increasing reproducibility of scientific results.

Minor comments

Point 1

The authors should re-read their supplementary material as there are some grammatical errors (e.g., was be made).  Additionally, some figures are missing (Figure S1 and Figure S5).

Response 1: We have corrected the grammatical errors and confirmed that the figures are all visible in the revised pdf file.

Point 2

The scale bars of the TEM figures are barely visible.  The authors should change either the color of the bars or the weight of the stroke.  Also, they do not need to write the value of the bar on the figure, since it can be barely seen.

Response 2: The scale bars were altered in all figures, as suggested.

Point 3    

The authors should check the numbering of the heading.  The heading 3.6 should be followed by the heading 3.6.1 and not 3.1.1 (line 475, page 15).

Response 3: This has been corrected.

Point 4

The authors should explain why someone should use the switchSENSE method and why did they use it only once in their experiments.

Response 4: SwitchSENSE is a biophysical characterization method ideally suited for analysing the binding kinetics of challenging molecules such as aggregation-prone proteins. The low surface density of the immobilized protein of interest allows controlled induction of oligomerization processes upon analyte injection in the liquid phase. switchSENSE has been applied here to determine the homo-dimerization association and dissociation rate constants of the full-length wild-type Ataxin-3, as well as the JD domain alone. These data indicate the concentration range associated with the onset of JD-dependent self-assembly and hint at the stabilizing role of the different protein domains in the aggregation process. It is therefore experimental support of the previously determined computational aggregation model and an orthogonal technique to establish the optimal assay conditions for the here presented methods of aggregation observation. We included a brief description of the advantages of this biophysical technique in the revised version of the manuscript (lines 261-269, page 6).

Reviewer 2 Report

This is an excellent, timely and important topic in the field. Find the manuscript well reasoned, thought-out and presented. The information regarding specific conditions that impact the aggregation of atxn3 protein in vitro will be helpful to the SCA3/MJD field, and likely also beyond, as these data also set the stage for proper, downstream analysis of compounds or conditions that than help prevent , or reverse, aggregation in vivo, for therapeutic purposes. 

I only have one small question that perhaps the authors can address textually: how did they decide to use Ub3-K63 chains, specifically, for their assays? 

Author Response

Point 1

I only have one small question that perhaps the authors can address textually: how did they decide to use Ub3-K63 chains, specifically, for their assays? “

Response 1: We thank the reviewer for the time dedicated to read our manuscript and for the overall comments on the work presented.

Concerning the question about the use of Ub3-K63 chains, we would like to clarify that in this work we use linear Ub3 chains. In these constructs the Ub chains are connected head-to-tail with a peptide bond linking the C-terminal of one Ub and the N-terminal of the following Ub. Although the effect of Ub on the aggregation of the Josephin domain of Ataxin-3 had been previously reported (Masino et al, 2010, doi: 10.1096/fj.10-161208), there were no studies showing how polyUb chains interfere with the aggregation of the full-length protein. Since ataxin-3 preferably interacts with polyubiquitin chains and cleaves K63-linked Ub chains (Winborn et al 2008, doi: 10.1074/jbc.M803692200), we chose to evaluate the effect of linear Ub chains on the aggregation of normal and polyQ-expanded ataxin-3. The reason for using these linear Ub chains is now briefly described on page 17 (lines 577-583) of the revised version of the manuscript.

Round 2

Reviewer 1 Report

The authors have satisfactorily addressed most of my concerns.  In particular, the authors changed the title and the manuscript, highlighting the main points of their work. 

I only have to point out that two figures are missing in the supplementary material (Figures S1 and S5), probably due to the conversion of the original file to pdf, which should be added.

Author Response

We thank the reviewer for the comments and will confirm that all the figures are visible in the final version of the supplementary material. A complete spell check was performed.